# Exposure of Von Willebrand Factor Cleavage Site in A1A2A3-Fragment under Extreme Hydrodynamic Shear

**DOI:** 10.3390/polym13223912

**Published:** 2021-11-12

**Authors:** Olivier Languin-Cattoën, Emeline Laborie, Daria O. Yurkova, Simone Melchionna, Philippe Derreumaux, Aleksey V. Belyaev, Fabio Sterpone

**Affiliations:** 1Laboratoire de Biochimie Théorique, CNRS, Université de Paris, UPR 9080, 13 rue Pierre et Marie Curie, F-75005 Paris, France; languin@ibpc.fr (O.L.-C.); laborie@ibpc.fr (E.L.); philippe.derreumaux@ibpc.fr (P.D.); 2Faculty of Physics, Lomonosov Moscow State University, 119991 Moscow, Russia; iurkova.do16@physics.msu.ru; 3Dipartimento di Fisica, Università Sapienza, P.le A. Moro 5, 00185 Rome, Italy; simone.melchionna@gmail.com

**Keywords:** von Willebrand factor, molecular dynamics, coarse-grains, lattice Boltzmann, shear flow, protein unfolding

## Abstract

Von Willebrand Factor (vWf) is a giant multimeric extracellular blood plasma involved in hemostasis. In this work we present multi-scale simulations of its three-domains fragment A1A2A3. These three domains are essential for the functional regulation of vWf. Namely the A2 domain hosts the site where the protease ADAMTS13 cleavages the multimeric vWf allowing for its length control that prevents thrombotic conditions. The exposure of the cleavage site follows the elongation/unfolding of the domain that is caused by an increased shear stress in blood. By deploying Lattice Boltzmann molecular dynamics simulations based on the OPEP coarse-grained model for proteins, we investigated at molecular level the unfolding of the A2 domain under the action of a perturbing shear flow. We described the structural steps of this unfolding that mainly concerns the β-strand structures of the domain, and we compared the process occurring under shear with that produced by the action of a directional pulling force, a typical condition of single molecule experiments. We observe, that under the action of shear flow, the competition among the elongational and rotational components of the fluid field leads to a complex behaviour of the domain, where elongated structures can be followed by partially collapsed melted globule structures with a very different degree of exposure of the cleavage site. Our simulations pose the base for the development of a multi-scale in-silico description of vWf dynamics and functionality in physiological conditions, including high resolution details for molecular relevant events, e.g., the binding to platelets and collagen during coagulation or thrombosis.

## 1. Introduction

Von Willebrand Factor (vWf) is a giant extracellular blood plasma protein that plays a key role in arterial hemostasis and thrombosis. Normally, in vivo vWf is a linear multimer consisting of 40–200 covalently linked monomers [1]. With a length of 60 to 80 nm, a monomer comprises 2050 amino acids distributed between multiple domains, each a few nanometers in size. One of the essential features of vWf is that these proteins can expand in the bloodstream [2,3,4,5] and capture the blood platelets in hydrodynamically adverse conditions, i.e., under extreme forces and mechanical stresses [6]. Vascular trauma, wound, plaque rupture, or inflammation are rapidly followed by the vWf-mediated adhesion and aggregation of blood platelets at the surface of damaged endothelium or exposed collagen [7,8,9,10]. After a mechanical stimulus (e.g., elevated shear, elongational stress, attachment to surface) the vWf multimers change shape from compact to extended, providing adhesive sites for platelets [4,11,12]. The stretching forces can lead to a number of conformational changes within the domains of this protein [13,14]. It is known that the deficiency of vWf concentration in blood, as well as the short length of these macromolecules, causes bleeding disorder [15]. At the same time, ultra-long vWf concatamers may cause thrombotic conditions [16].

The three neighbouring globular domains A1, A2 and A3 attract the most attention as they provide important properties related to platelet adhesion and aggregation. A3 domain is responsible for binding to collagen and thus initiation of thrombosis. A1 domain can establish adhesive bonds with GPIb platelet membrane receptors and is crucial for platelet plug formation under high shear stress [10,17]. Mechanical stability of A2 domain is essential for size control of vWf multimers.

During primary hemostasis in arteries and arterioles, platelet adhesion depends on the length distribution of the vWf multimers [18,19,20]. The globular A2 domain (unlike A1 and A3) does not contain disulfide bonds between its N- and C-terminal ends [21]. This fact makes vWf-A2 remarkably susceptible to mechanical tension [14]. Control of the multimer sizes in vivo occurs owing to this structural feature: after a forced unfolding of A2 domain and exposure of the scissile bond located between the residues Tyr1605-Met1606, the protein can be cleaved by the metalloprotease ADAMTS13 [22,23]. Such force-induced proteolysis, as observed for the ADAMTS13-vWf system, represents a model for probing the molecular mechanisms underlying the translation of a mechanical stimulus into a chemical response in a biological system [24]. The basic principle of this mechanoenzymatic phenomenon relies on hydrodynamic forces and torques exerted on vWf molecules in a sheared viscous fluid. The evolutionary developed regulation of vWf activity is extremely important for thrombosis and normal hemostasis [16,25]. However, its response depends on protein sequence, globular structure and stability of both the enzyme (ADAMTS13) and the substrate (vWf-A2). The outcomes of a inherited or acquired failure of the vWf activity can lead to severe or even life-threatening consequences [26].

Mechanical stability of A2 domain is thus one of the major points of interest with respect to vWf thrombogenic activity and regulation. Characterization of the mechanical properties of vWf at the molecular scale is important for understanding its hemostatic functions. Prior studies underline the resistive role of the central β-sheet and hydrogen bond networks in case of steered Molecular Dynamics (MD) simulations within the constant loading rate regime [27]. Structural analysis also suggests that calcium ions stabilize the native conformation of vWf-A2 domain [28]. However, experiments with optical tweezers showed that the calcium rather accelerates the refolding, without affecting mechanical unfolding of the globule [29]. Several recent works are devoted to the stability and interactions of C-domains, C-terminal dimeric bouquets, D’D3-, and D4-domains under high and even pathological mechanical load [30,31,32,33,34]. Many theoretical studies (steered MD simulations [27,35,36]) and experimental setups (Atomic Force Microscopy [34,37,38], Optical and Magnetic Tweezers [14,33]) were designed to understand the biomechanical basis for vWf properties by reproducing the unidirectional pulling. In other cases, the effect of shearing was considered by means of extremely coarse-grained simulations [12,39,40], as well as experimentally [3,4,11].

Further insights emphasize the importance of the structure of globular domains and flexible linker segments for understanding biogenesis, shear-induced conformational changes, platelet recruitment abilities, and mechano-chemical regulation of von Willebrand factor [30,41,42]. There is a growing evidence that relative spatial arrangement of neighbouring domains of vWf may lead to substantial functional changes of this protein. Recent studies revealed that interactions between A1 and A2 domains are critical for the attenuation of vWf adhesivity to platelets [35]. On the other hand, in case of adhesion to collagen, the three-domain constructs A1A2A3 may compensate for clinically relevant point mutations in the A3 domains [37]. Several experimental works used the A1A2A3 fragment as an elementary molecular model for vWf functioning, including its proteolysis by ADAMTS13 [38,43]. Therefore, not just the unfolding dynamics of individual A2 domains, but the collective behaviour of A1A2A3 constructs is likely to be involved into the regulation of the hemostatic activity of vWf. The detailed understanding of the exact A2-unfolding pathway and scissile bond exposure under different regimes of mechanical loading (pulling or shearing) is still missing in cases of A1A2A3 constructs and whole vWf multimers.

In the present work we use computer simulations to study molecular details of the A2 unfolding and the exposure of the scissile bond Tyr1605-Met1606 (or, using the indexing that begins from the N-terminus of A2 domain, ${}^{111}$Tyr-${}^{112}$Met). We use the Lattice Boltzmann Molecular Dynamics technique [44] based on the coarse-grained (CG) model OPEP [45]. We first validate the capability of the OPEP force field to reproduce the mechanical unfolding of the A2 domain by comparing CG and all-atom simulations [46]. Then, the shear induced unfolding of the A2 domain in presence of its natural molecular surrounding is studied by modelling of the whole A1A2A3 fragment. We focus on the hydrodynamic consequences of the attachment of A1 and A3 domains to the A2 domain.

The manuscript is organized as follows. We describe the simulation methods and analysis techniques in the Materials and Methods section. The Results section presents and analyses the results concerning the force and shear induced unfolding of the individual A2 domain, and the shear induced unfolding of the A2 domain in the A1A2A3 construct. A final discussion in presented in the Conclusions.

## 2. Materials and Methods

In this section we describe the model and the methodology used through this work. We first present the flexible coarse-grained model OPEP that we exploited to investigate the unfolding of the A2 domain under tensile force and shear, and the simplified representations introduced to model the elementary three-domains construct A1A2A3 simulated in shear flow. A pictorial representation of the A1A2A3 system is given in Figure 1. We then describe the Lattice Boltzmann Molecular Dynamics (LBMD) technique that we employed to explore the shear-induced unfolding process. Finally, we detail the theoretical models used to extract the unfolding kinetics at different force and shear regimes.

### 2.1. Protein Structures

In order to construct our systems we used as starting point the crystallographic structures of the three domains A1 (PDB code 1AUQ [47], 207 residues), A2 (PDB code 3GXB [22], 177 residues), and A3 (PDB code 1AO3 [48], 187 residues), see Figure 1a. In order to model the initial configuration of the flexible linkers connecting A1 and A2, and A2 and A3, we generated a linear configuration of the amino-acid chain then relaxed using the OPEP force field as described in the following, see Figure 1b.

### 2.2. The OPEP Model

The OPEP force field is a multi-resolution coarse-grained model developed to investigate peptide and protein folding without ad-hoc biases, and aggregation [45,49,50]. Namely, small proteins were successfully folded using enhanced sampling simulations based on the OPEP force field [45,50], the experimental different thermal-stability of homologous proteins was successfully reproduced [51], and the mechanical unfolding of small proteins under pulling forces compares very well with the process simulated using an all-atom force field [46]. OPEP is based on the atomistic resolution of the backbone while each amino acid side chain is represented by a single bead (see Figure 1b). Glycine and proline are exceptions and are fully modeled at atomic resolution. The Hamiltonian is composed of two sets of potential energy terms: the bonded terms, that ensure molecular topology as well as preferential dihedral orientation in the backbone, and the non-bonded terms, that implicitly include screened electrostatic interactions, so that all non-bonded interactions are short-range in nature. Specific hydrogen bond cooperative terms are included to favor the formation of secondary structures. A detailed description of the Hamiltonian has been presented in previous works, see ref. [45]. In the simulations presented hereafter we used the force field version v.4 [45]. The use of a fully flexible force field such as OPEP is mandatory for describing folding/unfolding or large conformational changes but is not necessary when the main conformation of the protein is preserved in time. In this latter case in order to increase the computational efficiency, it is preferable to exploit an elastic network model based on the OPEP force field for the intermolecular non-bonded interactions, and a reduced resolution. The amino acids are then represented using only the alpha carbons ($C_{\alpha }$) of the backbone and the side-chain beads. All neighbouring pairs of particles within a given spatial cut-off (in our case 6 Å) are linked by a network of elastic potentials. For each pair of interacting sites, the equilibrium distance of the elastic potential is set to the inter-site distance found in the crystallographic native configuration, and the spring constant is set to 5 kcal/(mol Å${}^{2}$) to ensure sufficient internal rigidity. This elastic representation was implemented to model the A1 and A3 domains since they remain stable under vWf elongation.

Finally, in order to model the linkers between domains A1 and A2 (L${}_{12}$), as well as A2 and A3 (L${}_{23}$), we derived a simplified chain representation based on the OPEP force field. For each linker we first performed a simulation with the full flexible OPEP force field. In this modelling we did not account for the glycosylations of the amino-acids. By focusing on the $C_{\alpha }$ positions, we extracted the overall bond and angle distributions for three consecutive $C_{\alpha }$ atoms, and we derived the equilibrium mean values as well as the force constants of the associated harmonic potentials via Boltzmann inversion, see Appendix A. The obtained values are, $r_{0}$ = 3.8 Å and $k_{b}$ = 149 kcal/(mol Å${}^{2}$) for the bond potential, and $\theta_{0}$ = 110${}^{{}^{\circ}}$ and $k_{\theta }$ = 1.5 × 10${}^{-3}$ kcal/(mol) for the angular potential. A torsional potential based on typical OPEP backbone parameters ($\phi_{0}$ = 180${}^{{}^{\circ}}$, multiplicity n=2 and force constant $k_{\phi }$ = 1 kcal/mol) was also introduced to ensure a correct extension of the chain. Finally, in order to provide a chemical flavor to the simplified representation of the linkers, we added a non-bonded potential between the beads. First, we classified the amino acids in two main groups, polar (P) and hydrophobic (H) ones, then we employed the OPEP potential interactions parameters to obtain an averaged interaction potential for the interactions of P-P, H-H, and P-H beads.

### 2.3. The Lattice Boltzmann Molecular Dynamics Framework

The LBMD, or in a more general term Lattice Boltzmann particle dynamics [44], has been previously introduced in the context of polymer physics to incorporate hydrodynamic effects in simulations based on implicit-solvent molecular representations [52,53,54]. The approach has also recently been extended to the investigation of biological systems at quasi-atomistic resolution [55,56]. In particular, the coupling between particle and fluid dynamics was effective in modeling amyloid aggregation [57,58], crowded protein solutions [45,59], nanoscale vesicles [60], and protein unfolding under shear flow [46,61].

The coupling between particles and solvent arises from a Stokes-like drag force acting on each particle:(1)$${\vec{F}}_{i}^{D}=-\gamma \left({\vec{v}}_{i}-{\vec{\tilde{u}}}_{i}\right)$$
where ${\vec{v}}_{i}$ is the *i*-th particle’s velocity, ${\vec{\tilde{u}}}_{i}$ is the fluid velocity $\vec{u}$ smeared over a finite extension of the *i*-th particle, and γ is the frictional coupling, an adjustable parameter in the methodology. The drag force adds up to the usual conservative forces derived from the Hamiltonian of the system, ${\vec{F}}_{i}^{C}=-{\vec{\nabla }}_{i}U\left(\left\{r\right\}\right)$ and to a random white noise, ${\vec{F}}_{i}^{R}$, that represents thermal fluctuations.

In our simulations, the LB implementation uses the BGK (Bhatnagar-Gross-Krook) collisional operator [62] with a lattice spacing of 5 Å, a resolution needed to sufficiently resolve local hydrodynamic interactions for macromolecular systems. The solvent kinematic viscosity was set to the value for bulk water at ambient conditions.

For the dynamics of the A1A2A3 system we used a multiple time stepping to separate the integration of the bonded interactions involved in the flexible A2 domain, the A1/A3 elastic network domains and L${}_{12}$/L${}_{23}$ linkers (timestep of 3 fs), and the non-bonded interactions (timestep of 9 fs). The molecular and fluid dynamics were evolved synchronously at the largest molecular timestep. The LBMD simulations were carried out using the code MUPHY [63]. Technical aspects concerning the method and its numerical setup as well as shear generation have been detailed in previous works [55,58,61]. The solvent undergoes an external perturbation and produces a Couette flow. This, in turn, is perturbed by the presence of the vWF protein and its dynamics (two-way exchange). This modifies locally the linear velocity gradient of the Couette flow (see Figure 1c and Appendix A). In order to avoid finite size effects along the elongation direction we set up a simulation box of size 80×20×20 nm.

### 2.4. All-Atom Molecular Dynamics

All-atom simulations of clamp-force unfolding of the A2 domain were carried out in Gromacs 2018.4 patched with PLUMED 2.5, using the Amber a99SB-disp force-field for disordered and ordered proteins together with its TIP4P-disp water model [64]. The structure of the A2 domain (PDB:3gxb) was processed using the Gromacs package tools and solvated in a ∼7×7×50 nm simulation box with 10 Na${}^{+}$ ions to ensure charge neutrality. The system was minimized then equilibrated in the NPT ensemble (300 K, 1 bar) using the canonical velocity-rescaling thermostat and the Parrinello-Rahman barostat. In order to obtain uncorrelated unfolding trajectories, five 10 ns equilibration trajectories were produced using different random seeds and snapshots at 6, 7, 8, 9 and 10 ns were extracted from each one in order to generate 25 starting configurations for the following SMD experiments. The end-to-end distance projection on the z-axis was biased with a constant force using PLUMED. In all cases the protein very fastly aligned along the pulling direction. An upper wall was set at 46 nm to prevent extension passed the box size. Simulations were run from 20 ns up to 180 ns so that a majority achieved unfolding of the cleavage site. The z-projected end-to-end distance was recorded every 10 ps and unfolding kinetics were analyzed using the Maximum Likelihood (ML) framework described bellow.

### 2.5. Kinetic Models for Unfolding

The unfolding kinetics of a protein subjected to a pulling tensile force has been often described using a two-state irreversible Markovian model for the underlying evolution. In this model, the first passage time τ to move from the folded configuration toward the unfolded one follows an exponential probability distribution that depends on the forward reaction rate *k*:(2)f(τ;k)=kexp(−kτ)

According to Bell’s model [65], when an external force is applied to one terminus of the protein, the barrier-limited unfolding transition is accelerated because of the added mechanical work along the unfolding path, $k=k_{0}e^{\beta F\delta x}$, where $k_{0}$ is the rate at null force, $\beta =1/k_{B}T$, and δx is the distance between the reactant state and the transition state along the pulling direction. This expression is valid for a two-state, thermally activated reaction happening on a one-dimensional energy landscape, conditions that can be challenged in realistic systems [66]. Moreover, in MD simulations that generally allow to access molecular processes at the ns−μs timescale, computational limitations impose the use of pulling forces higher than those usually employed in experiments. In the high-force, diffusion-dominated regime, one can instead model the unfolding process as a Brownian motion with drift along the unfolding coordinate [67]. In that case, the time τ for going from the reactant state (at x=0) to some unfolded state (at x=α) follows an inverse Gaussian distribution that reads as follows:(3)$$f\left(\tau ;\mu ,\lambda \right)=\sqrt{\frac{\lambda }{2\pi \tau^{3}}}\exp\left(-\frac{\lambda {\left(\tau -\mu \right)}^{2}}{2\mu^{2}\tau }\right)$$
with the parameters of the distribution defined as μ=α/ν and $\lambda =\alpha^{2}/\sigma^{2}$. They physically relate to the displacement amplitude of the underlying Brownian motion σ, and to the drift ν that incorporates the combined effect of the external force and of diffusivity, i.e., the action of friction along the path.

### 2.6. Maximum Likelihood Analysis of Unfolding Kinetics

In order to characterize the unfolding dynamics of the A2 domain under the action of an external pulling force, or fluid shear, we followed the time evolution of several independent runs and introduced a suitable order parameter to describe the time-dependent unfolding. At a given threshold of the order parameter, an unfolding event is recorded, yielding for that simulation a first unfolding time τ. In cases where the unfolding criterion was not reached before the end of a trajectory, the final time $\tau^{c}$ was registered instead. Finally, using the extracted distributions of unfolding times or simulation times we estimate the parameters of the two kinetic models described above using a maximum likelihood (ML) framework for survival analysis. Let f(τ;θ) be the probability density function for first unfolding time τ in a given model (exponential model or inverse gaussian model), with parameters θ. Let ${\left\{\tau_{i}\right\}}_{i\in 1...N}$ be the sample of observed unfolding times and ${\left\{\tau_{j}^{c}\right\}}_{j\in 1...M}$ the ending times of trajectories that did not unfold yet (that is right censored data). The likelihood function is defined as:(4)$$L\left(\theta \right)=\prod_{i=1}^{N}f\left(\tau_{i};\theta \right)\prod_{j=1}^{M}\int_{\tau_{j}^{c}}^{+\infty }d\tau f\left(\tau ;\theta \right)$$

The best estimate $\hat{\theta }$ for the model parameters is then given by:(5)$$\hat{\theta }=argmax_{\theta }L\left(\theta \right)=argmax_{\theta }\ln L\left(\theta \right)$$
where the logarithm of the likelihood function is used for computational convenience.

For the exponential model (Equation 2), each force is analyzed independently using the corresponding dataset ($\theta \equiv k_{F}$). For the inverse Gaussian model (Equation 3), the scale parameter λ is estimated jointly for all forces while the location parameter μ is estimated for each of the *n* forces ($\theta \equiv \{\mu_{F_{1}};\mu_{F_{2}};...;\mu_{F_{n}};\lambda \}$).

The analysis of the simulation data was performed using the Python scripting language. The parameters were estimated using the *minimize* function of the SciPy package.

## 3. Results

### 3.1. Validating the Unfolding of the A2 Domain

In this section we discuss the unfolding of the A2 domain under the action of an external mechanical force. We compare the results from CG and all-atom simulations. The comparison of the unfolding kinetics and structural unfolding paths between the two sets of simulations is presented as validation of the CG model for the study of the extended A1A2A3 fragment under shear flow.

#### 3.1.1. Single A2 Domain: Pulling with a Constant Force

We carried out steered Molecular Dynamics simulations at 300 K on the A2 domain of vWf modelled using the OPEP force-field. Coupling with the fluid was not considered during these simulations. The domain was simulated in a box of dimension 14×7×7 nm. In the simulation a constant force is applied to one terminus of the domain while the other is kept fixed in space during the dynamics. We explored a range of forces between 300 pN and 750 pN. Generally, the simulations were stopped once unfolding was achieved. Otherwise, we extended the trajectories up to 100–500 ns, depending on the force value. For each value of the pulling force, up to 20 independent simulations were performed. We also performed a set of atomistic simulations used as a basis of comparison for the unfolding mechanism. These MD simulations were based on the Amber99SB-disp force field [64], and using pulling forces of 400 pN and 600 pN.

In order to quantify the accessibility of the cleavage site for ADAMTS13, we define unfolding as the full exposure of the core amino acid ${}^{111}$Tyr. We further define a collective variable counting the number of backbone atoms within 10 Å of any of ${}^{111}$Tyr atoms (excluded), this variable measures how the residue is buriedin the protein matrix before getting exposed to the solvent, and referred as coordination number. This definition allows to compare the results from coarse-grained and all atom simulations. When plotted against time (see left panels in Figure 2), the coordination number of ${}^{111}$Tyr decreases as parts of A2 unfold and the tyrosine gets exposed to the solvent. Along a typical unfolding trajectory, in agreement with past studies [36], the domain visits several metastable states, whose lifetimes increase when the pulling force is lowered. It should be noted that for the lowest tested force, 300 pN, none of the trajectories displayed significant exposure of ${}^{111}$Tyr on the simulation timescale.

When the number of neighbouring atoms is under 50, we consider ${}^{111}$Tyr fully exposed and the protein unfolded. Since unfolding is a stochastic process, some trajectories, especially at low force values, did not reach this criterion before the end of the simulation. While in these cases we cannot devise a first unfolding time, they are nonetheless informative since they yield a lower bound for it. Hence, we use a maximum-likelihood approach to analyze together both the “unfolded” trajectories and the “not unfolded” ones (see Methods). For each force value *F* the best estimate for the unfolding rate $\hat{k}\left(F\right)$ is expressed as the inverse of the mean first unfolding time, and is computed for the two kinetic models described in the Method section. For the exponential model, each set of simulations for a given force value *F* is analyzed independently to evaluate $\hat{k}\left(F\right)$. For the inverse Gaussian model we could not proceed with an independent evaluation of both $\hat{\mu }\left(F\right)$ and $\hat{\lambda }\left(F\right)$ at each force given our sample sizes. Instead, we further constrained the optimization problem by assuming a force-independent $\hat{\lambda }$. Such an assumption is consistent with the interpretation that λ relates to the elementary displacement of the underlying Brownian motion (σ) and the escaping distance (α), none of which should depend on the applied force. On the other hand, independent parameters μ(F) are provided for each force value *F*. The resulting set of parameters was optimized globally using a single ML procedure. The obtained results are shown in Figure 2. The central chart shows the unfolding kinetics obtained using the two models. The panels in the right part of the figure report the unfolding time distributions associated with the two models. The results from atomistic simulations are also shown for comparison and were fitted with the exponential model only, since the inverse Gaussian model could not be reliably fitted on this limited dataset.

The comparison between the distribution shapes and the empirical data strongly suggests that exponential decay does not apply well to the high force regime investigated, 400–750 pN. A lower force of 300 pN was also simulated but hardly reached the unfolding criterion on the simulation time frame. In this regard, the inverse Gaussian model offers much more realistic predictions. Moreover, in the exponential fit, estimated rates strongly deviate from the exponential dependence on applied force predicted by Bell’s model. Taken together, the results suggest that the dynamics of the system is highly diffusive in this high-force regime and that the observed kinetics cannot be simply related to a phenomenological energy barrier extracted from lower-force experiments. Hence, the observed dependence on force does not arise from easier crossing of a single well-defined free-energy barrier, but can be understood as increased drift velocity in the driven Brownian model. This agrees to what already observed in other molecular simulations, where the A2 mechanical unfolding is a multi-step process proceeding via several metastable states [36].

#### 3.1.2. Single A2 Domain: Structural Analysis of Force Unfolding

In order to characterize the progress of the unfolding that leads to the exposure of ${}^{111}$Tyr, we visually inspected the trajectories and singled out three sequential structural events: (i) the unzipping of β strands 5 (res. 129–136) and 6 (res. 155–157); (ii) the unzipping of β strands 4 (res. 108–114) and 5; and (iii) the unzipping of β strands 1 (res. 3–10) and 4. We note that the unfolding of the A2 domain in the force-clamp setup begins from the C-terminus and demonstrates transient intermediate states in accordance with prior experimental and structural studies [14,22]. A pictorial representation of this sequence is given in Figure 3, left panels.

To support that intuition we analyzed the number of native hydrogen bonds formed between each β strands pair and involving the backbone atoms. To this purpose, native contacts between the backbone O and H${}_{N}$ atoms are defined using a 3 Å cutoff in the reference crystal structure. Then, the number of contacts is defined as:(6)$$C\left(t\right)=\sum_{contacts}\frac{1}{1+\exp\{a\left(d\left(t\right)-d_{ref}\right)\}}$$
where *a* is a smoothing factor (set at 0.5 Å^−1^), d(t) is the O-H${}_{N}$ distance at time *t* and $d_{ref}$ is the distance in the reference structure.

The number of contacts for the three pairs of β strands are accumulated for all trajectories at a given force and shown on Figure 3 as 3D scatter plots. For the sake of comparison we have reported the results from OPEP CG simulations and all atoms simulations. The sequential breaking of each part of the β sheet is very clear for all-atom simulations. While OPEP exhibits a more flexible behaviour, the overall sequence of event is respected on average. Trajectories that do not reach full unfolding stay in metastable states either at the second (β 4–5) or the third (β 1–4) steps, hinting for the key role of these structural motifs in the mechanical resistance of A2.

This sequence of unfolding detected in our simulations is consistent with that described in previous work using SMD simulations [27,36].

#### 3.1.3. Exposure of the Cleavage Site

It is worth noticing that the cleavage of vWf-A2 is a multistage process that obeys the known “key-lock” principle. The active domain of metalloproteinase ADAMTS13 has relatively low proteolytic activity against human von Willebrand factor, and they are sometimes considered as a non-optimal enzyme-substrate pair [68]. Before the metal ion cuts the bond, the enzyme must anchor to the unfolded vWf, recognize correctly the location of the cleavage site and place the active site properly [43]. The exposure of the cleavage site is a necessary, but not a sufficient condition.

In the disintegrin-like domain of ADAMTS13 a disordered region called V-loop is present. Although being variable in length and sequence, such region is also found in different ADAM-family proteins [69]. In ADAMTS13 this region contains certain charged residues located near Leu350 and Arg349, which should interact with a few specific residues, most probably Ala1612 and Asp1614 (${}^{118}$Ala and ${}^{120}$Asp in the local notation), of the unfolded A2 domain [24,70,71]. Recently, it was suggested that these charged residues may collaboratively create a vWf-binding exosite on the surface of the metalloprotease [24]. This hypothetically allows to position the scissile bond Tyr1605-Met1606 for cleavage, noticeably affecting the rate constant and catalytic efficacy of proteolysis [43]. Given the distance between the aforementioned residues in ADAMTS13, the uncoiled part of A2 should not only contain the scissile bond itself, but also a segment of approximately 25 Å in the direction to the C-terminal end, which we here refer to as the *recognition* site. In case of force-pulling since the force is applied at the C-terminus the recognition site is exposed before the cleavage site, see Figure 4. As mentioned above, under real conditions, the unfolding of the A2 domain is however triggered by the shear forces. Hence the unfolding directionality and the associated sequence of events might differ. For a set of unfolding simulations of A2 under shear flow we have monitored the exposure of the recognition and cleavage sites, see Appendix A. It appears that the correlation among the two sites exposures is lost, and less evident than in the force unfolding. This is somehow not surprising since the isolated domain rotates in the shear field and different structural elements are periodically exposed to the elongation component of the flow that causes the unfolding to proceed along a variety of pathways. This is further confirmed by looking at the unfolding of the elementary secondary structures as reported in Appendix A. The variability of unfolding pathways under shear flow when compared to the unfolding caused by a directional pulling force was already observed for other proteins, and relates to the cyclic application of the elongation force to different portions of the unfolding protein structure [46,61].

### 3.2. The Three Domains A1A2A3: The Unfolding of A2 in Shear Flow

In the previous section we have shown that the OPEP model describes the force induced unfolding paths and kinetics of the isolated A2 domain very similarly to what obtained by an all-atom model in explicit solvent. Here, we investigate a more realistic system, with the A2 domain linked to the neighbouring domains, A1 and A3, as found in the vWf. The complete system is described in the Methods section. In order to study the effect of shear flow we first generated a set of independent initial states by equilibrating the system in the absence of shear perturbation and by extracting a configuration every 1 ns. In these initial configurations the A1 and A3 domains are not in contact with A2, but the linkers are not fully extended. Each configuration was used to start a simulation where the fluid shear flow was activated. First we considered the high shear regime that induces unfolding and exposure of the cleavage site on a timescale of tenths of nanoseconds.

The exposure dynamics of ${}^{111}$Tyr are monitored as described in the previous section by considering the decrease of the number of A2 domain backbone atoms contained in a sphere of radius $r_{c}=10$ Å centered at the cleavage site. The evolution of the coordination number of ${}^{111}$Tyr is reported in Figure 5 for several trajectories and for different shear rates. While at the highest shear rate almost all trajectories lead to the complete exposure of ${}^{111}$Tyr, weaker perturbations lead in many cases to incomplete unfolding and the cleavage site remains buried.

The elongation dynamics in shear flow can also be followed by monitoring the chain extension of the A2 domain expressed by its end-to-end distance, see Appendix A. We observe two typical scenarios: the first one, when a continuous diffusive-like evolution of the extension leads to unfolding and, therefore, to full exposure of the cleavage site (the contour length of A2 is ≃65 nm), and the second one, when the extension is limited by a metastable local configurational lock, and eventually reverses back. This latter type of event is caused by the rotational dynamics of the whole complex induced by the Couette velocity field and that leads to partial collapse of the three domains A1, A2 and A3. This compaction of the three domains reduces the tensile force acting on the A2 domain, which acquires an unfolded molten globule conformation instead of an extended linear one. In these cases the cleavage site ${}^{111}$Tyr is not accessible. A molecular view of the two scenarios is given in Figure 6 for complete exposure, and in Appendix A for partial exposure.

By using the same strategy adopted for the force-induced unfolding, we quantified the unfolding kinetics by applying the diffusive model (the most adequate to fit the data), see Figure 5. By comparing the characteristic unfolding time with those obtained for the simulation of A2 under force, we deduce a semi-quantitative match among the shear values 9–10 × 10^9^ s^−1^ and a directional mechanical force of 500–700 pN.

We have then verified the simultaneous exposure of the cleavage and recognition sites, see Figure 7. For the three domains system A1A2A3 the exposure of the recognition site anticipates slightly that of the cleavage site. One may extrapolate this observation on the vWf multimers: for longer chains the mechanical load on A2 domain is more resemblant to the directional pulling. This is especially pronounced for the grafted multimers, which usually experience greater local tension as compared to the free-flowing ones [12,72]. We also see that under a higher shear stress the extension of the A2 domain has a tendency to follow the complete unfolding pathway, rather than the reassembly pathway, so that the probability for the correct ADAMTS13-vWf complex formation is higher.

### 3.3. The Three Domains A1A2A3: The Shear-Induced Unfolding and Mechanical Tension

In this section we relate more explicitly the fluid shear flow to the mechanical tensile force that triggers the unfolding of the A2 domain. We recall that, at longer length scales, and considering specifically the multimeric vWf, the analysis of the tension profile recorded along the vWf chain modelled as a polymer moving under the action of shear flow and hydrodynamics interactions, suggested a general mechanism (the creation of a local protrusion) that leads to the transition from the collapsed to the elongated shape [40]. By combining scaling analysis with a two-state model that describes the A2 domain as open or close, it was proposed that the rate for ADAMTS13 cleavage activity is limited by the fraction of cleavage sites that get exposed because of the tensile action of the shear flow on the protrusions [39]. When focusing at the protein length scale, by using a simple mechanical model, and energetic considerations, Jaspe and Hagen [73] set the critical shear rate to unfold a mid-size globular protein to $\dot{\gamma }>10^{7}\;s^{-1}$. Numerical simulations based on the OPEP CG model confirmed this, and showed that in order to observe the unfolding of small/mid size proteins in the nanosecond timescale shear rates should be indeed as high as ∼$10^{9}\;s^{-1}$ [46,61]. At this shear rate value, the fluid acts on the rotating proteins applying an elongation mechanical drag along the unfolding reaction coordinate (e.g., the HB ramp in β-hairpin) that can spike at values in the order of several $10^{2}$ pN. This is coherent with the typical values for the forces applied in pulling MD simulations. Moreover, in the previous sections by comparing the kinetics of unfolding, we found an empirical correlation among the force and the shear rate required to unfold a protein in the nanosecond timescale [46].

We now quantify at the level of the A2 terminus, where the unfolding starts, how the action of the shear translates in a mechanical pulling force. In our model the terminal residues of the A2 domain are attached to the initial beads of the linkers L${}_{12}$ and L${}_{23}$ via a harmonic bond, $\frac{1}{2}k{\left(d-d_{0}\right)}^{2}$. Under the perturbation of the shear flow this bond is stretched, and it is possible to record the instantaneous stressing force $k(d\left(t\right)-d_{0})$. This stress can be averaged over a time interval $\tau_{w}$ in order to estimate the evolution of the local mechanical force acting on the A2 terminal sites along the unfolding process, $\bar{f}={\langle k\left(d\left(t\right)-d_{0}\right)\rangle }_{\tau_{w}}$.

In Figure 8 we show an exemplar plot, where for the shear rate $\dot{\gamma }=1.13\times 10^{9}\;s^{-1}$ we report the force acting on the A2 terminus via the L${}_{12}$ linker during the unfolding event. The force evolution is contrasted with the evolution of the coordination number of the ${}^{111}$Tyr. During a first initial transient phase of about 5 ns the force starts raising as effect of the fully elongation of the linkers that transfer the force to A2. In this phase the system is in tension and the first extension of the A2 domain occurs. After 10 ns the force peaks at a value of about 500 pN inducing a further extension of the domain. Such values agree with the estimations for local tension in multimeric vWf under shear or elongational flow [14]. The action of the force extends for the following 30 ns, with the tension progressively decreasing till a final extension jump that leads to the full exposure of ${}^{111}$Tyr.

In Appendix A we report similar plots for both the L${}_{12}$ and the L${}_{23}$ connections to the A2 domain obtained from several independent runs. We generally observe that the resulting pulling force is more intense at one edge than the other. This asymmetric response is due to the different lengths of the linkers, 37 and 15 respectively, and the different velocity field experienced by the attached domains. The longer linker L${}_{12}$ is more easily subject to tension and generates a loading force at the edge of the A2 domain higher that that generated at the other terminus by the shorter L${}_{23}$ linker. However, it must be noted, as also shown in Figure 6, that the unfolding of A2 proceeds at both C- and N-ter, and the protein is exposed to a total load that combines the pulling forces at the extremities.

### 3.4. The Three Domains A1A2A3: The Rotational Dynamics

In order to better characterize the competitive effects on the three domains of the extensional and rotational components of the shear flow, we built a second model of the A1A2A3 system. In this model, all three domains are described by an elastic network so that also the A2 domain maintains its folded structure. The model allows us to span longer timescales in the simulation since we can increase the integration time-step and reduce the intramolecular degrees of freedom for A2. Therefore we can investigate the overall rotational dynamics of the molecular construct, and the inter-domains interactions, even at small shear rate values (10^5^–10^6^ s^−1^).

First, we quantify the rotational dynamics of the two peripheral domains, A1 and A3, with respect to the central A2 domain. This is done by computing for each simulated system *i* the first order orientational time correlation function (TCF) $c\left(\tau \right)=<cos\left(\theta \left(\tau \right)\right){>}_{i}$, where cos(θ(τ)) is defined as the scalar product between two instances, separated by a lag time τ, of the vector connecting the centers of mass of A1(A3) and A2 domains, $\cos\left(\theta \left(\tau \right)\right)=R_{12(32)}\left(t+\tau \right)\cdot R_{12(32)}\left(t\right)$.

The obtained TCFs decay in time but their trend do not fit to standard decay models (e.g., exponential or stretched exponential). Therefore, from each TCF, we have individuated the value when it reaches zero the first time. These characteristic times averaged over independent trajectories and as a function of the shear rates are plotted in panel A and B of Figure 9 for the rotation of the two domains A1 and A3 with respect to the central one A2. The obtained results clearly show that below a critical shear rate of ∼$10^{8}\;s^{-1}$, the rotational dynamics of the peripheral domains is significantly slowed down (a factor 3–4 compared to the high shear values). In order to resolve such dynamics much longer simulations should be performed, but it seems that given the characteristic size of the A1A2A3 construct a shear threshold for fluid activated rotation exists. One would expect that for a longer multimer this threshold should downshift to a smaller shear rate value. Moreover, it is very intriguing that below the threshold the evolution of the relative rotation of this realistic molecular system deviates from the theoretical relationship between the field rotational period and shear rate value, $T_{r}=4\pi /\dot{\gamma }$, see Appendix A. The decoupling between external field rotational flow and molecular rotation is due to the different contribution of the thermal motion and fluid induced rotation, with the former dominating at low shear rates, $<10^{6}\;s^{-1}$.

The rotational dynamics implies that the molecular three-domain system is exposed cyclically, at least at the highest shear rates, to the tensile force and to angular acceleration. The angular velocity of the peripheral domains A1 and A3 with respect to the central A2 is computed by considering the variation of the vector connecting the centers of mass in the unit of time Δt, $\Delta R=R_{12(32)}\left(t+\Delta t\right)-R_{12(32)}\left(t\right)$, and having for small displacements the angular variation $\omega =\Delta R_{12(32)}/R_{12(32)}$. The angular velocity is then computed on the plane of the rotation (XZ), and reported as the average over time and replicas versus the shear rate in Figure 9, panel C. Again, we notice the activation of the domain relative rotation above a critical shear rate, ${\dot{\gamma }}_{c}∼10^{7}$–$10^{8}\;s^{-1}$. On panel D of the same figure, we report for A1 (top panel) and A3 (bottom panel) rotations the time evolution of the angular velocity for an individual trajectory and at two representative shear values, $1.4\times 10^{5}\;s^{-1}$ and $8.5\times 10^{8}\;s^{-1}$. When the rotation is activated by the shear flow at $\dot{\gamma }=8.5\times 10^{8}\;s^{-1}$, the angular velocity shows a typical cyclic pattern, with spikes periodically occurring every 30–40 ns. On the contrary for the lowest shear rate, the angular velocity fluctuates uniformly in time as result of the molecular thermal motion only.

In the same spirit of previous work [40], but at a very different scale, we inquire how the threshold individuated for the rotational dynamics, transposes in a neat transition of the A1A2A3 fragment conformational state. Namely, we quantified the propensity of the three domains to stay detached and elongated, or collapsed. The results are summarised in Figure 10 where we show the time-averaged value of the inter-domain distances as a function of the shear rate. It is clear that at low shear rate the three domains stay in a collapsed configuration and move as a compact unique entity. On the contrary, at high shear rate, the three domains extend cyclically (see also Appendix A). The individuated threshold coincides with the that of the rotational dynamics, ${\dot{\gamma }}_{c}∼10^{7}$–$10^{8}\;s^{-1}$. For a longer chain constituted by a larger number of domains, this threshold should decrease (although different models predict different scaling [40,73]). However, the structure of a vWf monomer [41], and the sequential linking of its domains, is expected to generate a hierarchical scale of detachment, elongation, and unfolding events. The collapsed structure observed at low shear rate, and also the cyclic collapse/extension dynamics typical of the high shear rate regime, point on the critical role of the relative domain interaction for self-inhibition toward the cleavage.

## 4. Discussion

In the present paper we used both full-atomistic and coarse-grained computer simulations to explore the unfolding of the A2 domain of vWf in two perturbative regimes typical of experimental setups: the AFM-like unidirectional pulling with a constant force, and the Couette flow typical for near-wall regimes of micro-fluidic flow chambers and rotational viscosimeters. We considered the isolated A2 domain and the A1A2A3 tri-domain construct of the vWf monomer. Our results show that the unfolding pathway of the A2 domain is controlled by the sequential disruption of the β-strands in the core of the globule, while the α-helices give almost no resistance to the deformation.

The unfolding kinetics under extremely large forces, according to the presented analysis, deviate from Bell’s law and are better approximated by the inverse Gaussian model. The exposure of the Tyr1605-Met1606 cleavage site may correspond to partial unfolding of the A2 domain. However, under stronger forcing, complete unfolding events leading to a fully linear unfolded protein structure have been observed in simulations over tens of nanoseconds timescale.

The unfolding pathway of the A2 domain depends on the deformation mode, applied forcing, and, importantly, on whether the individual A2 domain or the tri-domain A1A2A3 structure is loaded. For the A1A2A3 assembly in Couette flow the deformation pathway consists of the stretching of the A2 domain punctuated by relatively long periods of rotation-induced refolding, at least under high shear stress. Due to the tumbling motion, A2 can not only adopt a linear, extended conformation, but also a melted globule one that hinders ADAMTS13 binding and subsequent proteolysis. This contrasts with the behaviour of the individual A2 domain under hydrodynamic shear, in which we mostly detected transient periods of partial unfolding. Moreover, the results of CG simulations suggest that, if the shear stress is strong enough, the rotational component of the flow dominates over the protein’s Brownian motion above a critical threshold. In the shear flow we observed two different scenarios of A2 extension, revealing a complex interplay between fluid dynamics and molecular structure.

Several intriguing questions emerge, including the refolding pathway of the A2 domain after deformation, the reversibility of the unfolding, the structure of the compacted globule and influence of the protein’s molecular surroundings on this process. According to [14], refolding to the correct native state of the A2 domain when the tension is released is important for vWf functions in vivo, as an incorrectly refolded protein could become less resilient and eventually permit cleavage by ADAMTS13. We have observed that under a high shear rate in fluid, the A1A2A3 construct does not exhibit the native structure of A2 domain after the rotation-induced collapse. To understand what happens in a steady fluid or under a mild shear rate is a prominent goal for forthcoming research. Moreover, in our analysis we have focused on the complete exposure of the cleavage site as a prerequisite for ADAMTS13 intervention. Most studies generally assume complete unraveling of the A2 domain [74]. However, some uncertainty remains regarding the relevant A2 conformational state for cleavage. For example, Baldauf et al. [36] propose that the partially unfolded intermediate following β 5 unzipping might suffice for recognition and proteolysis.

Finally, by focusing on the relative inter-domain dynamics, we have individuated a shear-rate controlled dynamical configurational transition among a stable collapsed structure of the three domains (low shear) and a cyclically alternated collapsed/elongated structure (high shear). The preferential inter-domain interactions visible in the collapsed structures highlight the key role of these interactions in self-inhibition to cleavage.

The present study is a step toward a better understanding of the biophysical mechanisms that govern mechanoreception and mechanotransduction in proteins at the molecular level.

## Figures and Tables

**Figure 1 polymers-13-03912-f001:**
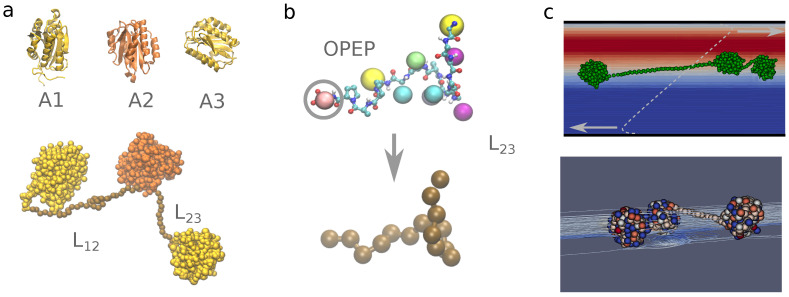
A1A2A3 system. Panel (**a**). Molecular representation of the A1A2A3 construct modelled in this work. In the top part we show the individual domains (from X-ray PDB structures) represented by their secondary structures. In the bottom part we show the A1A2A3 construct including the flexible amino acid sequences connecting the domains. Panel (**b**). The linker chain is represented using the OPEP (**top**) and a single bead (**bottom**) models for the amino acids. Panel (**c**). Representation of the Couette flow generated in the simulation box of the A1A2A3 system (**top**) with the velocity field represented by the colour gradient. In the (**bottom**) we highlight the local deformation of the velocity streamlines around the protein domains A1A2A3.

**Figure 2 polymers-13-03912-f002:**
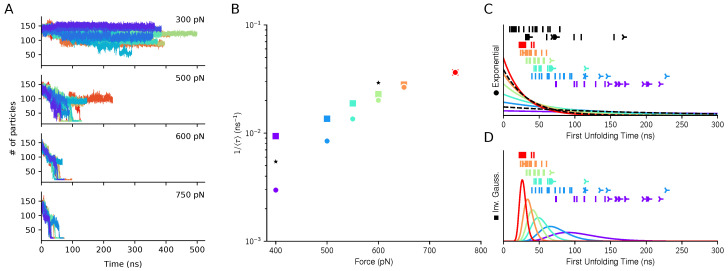
Exposure of the cleavage site upon force-clamp unfolding. Panel (**A**). Time evolution of the coordination number of ${}^{111}$Tyr under the action of mechanical forces of different magnitudes in coarse-grained simulations. At the beginning the residue is buried in the protein matrix, and the number of protein backbone atoms in its proximity is the highest. During unfolding this number decreases as a sign of progressive exposure out of the protein matrix. For each force value, up to 20 independent simulations were run. Panel (**B**). Kinetics for the cleavage site exposure. For the CG model the inverse mean first unfolding times is estimated for the exponential (circles) and inverse Gaussian (squares) distributions. For all-atom simulations only the exponential model is used (black stars). Panels (**C**,**D**). The estimated distributions are shown (solid lines) together with the simulated unfolding times (vertical bars) or end of not-yet-unfolded trajectories (three-pointed stars).

**Figure 3 polymers-13-03912-f003:**
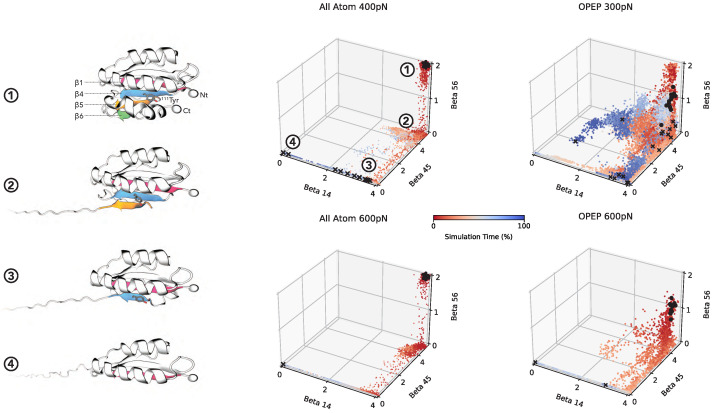
A2 unfolding path. (**Left**). From top to bottom, typical unfolding pathway of A2 in force-clamp simulation. As beta strands 6 (green), 5 (orange), 4 (blue) and 1 (red) unfold, residue ${}^{111}$Tyr is more and more exposed. Snapshots were rendered in VMD and numbered 1–4. (**Center**, **Right**). The unfolding trajectories are projected on the 3-dimensional CV space of the native H-bonds between beta strands 1 and 4, 4 and 5, 5 and 6. Black dots and crosses denote the beginning and ending of the trajectories, respectively. Trajectories are colored from red (beginning) to blue (ending). In the first chart, the number 1 to 4 indicate the typical protein structure along the unfolding path shown in the left of the figure.

**Figure 4 polymers-13-03912-f004:**
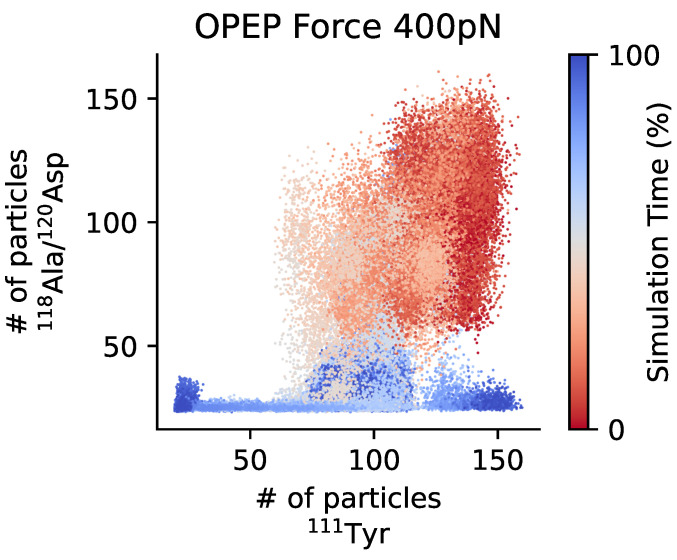
Exposure of the cleavage and the recognition sites of A2 domain under force. The CG trajectories of force unfolding (400 pN) are projected as a scatter plot on the 2D space of the coordination number of ^111^Tyr (cleavage site) and that of ^118^Ala/^120^Asp (recognition site). The sequential exposure of the recognition site before the cleavage site can be appreciated. Trajectories are colored according to progress from red (beginning) to blue (ending).

**Figure 5 polymers-13-03912-f005:**
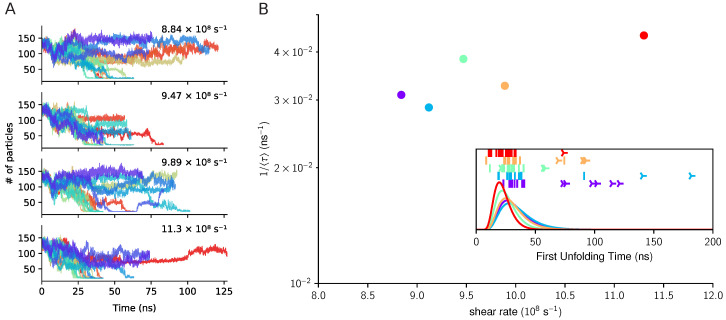
Exposure of cleavage site in A1A2A3 under shear flow. Panel (**A**). Time evolution of the coordination number of the cleavage site in A2 expressed as the number of backbone atoms inside a sphere of radius $r_{c}=10$ Å centered on ${}^{111}$Tyr. In the four panels we report data for several shear rates. Panel (**B**). Shear-dependent mean exposure kinetics derived from the estimate of the first unfolding time obtained from maximum likelihood approach (ML). In the inset chart the ML derived distributions for the exposure time are reported for the different shear rates (solid lines) together with the simulated unfolding times (vertical bars) or the end of not-yet-unfolded trajectories (three-pointed stars).

**Figure 6 polymers-13-03912-f006:**
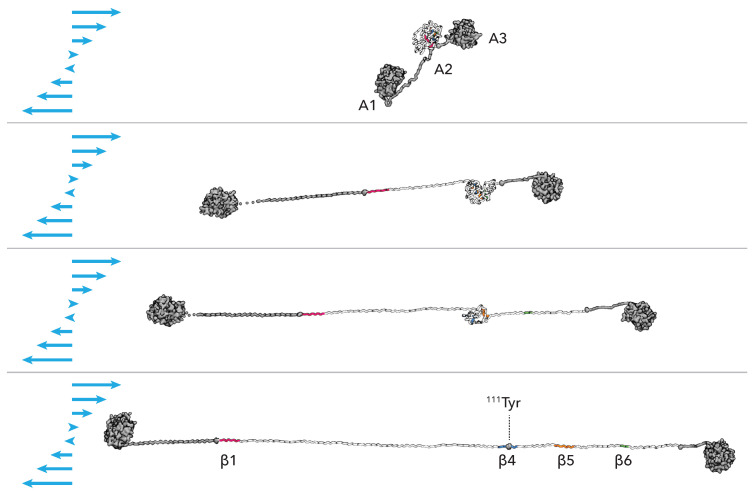
Molecular view of the A1A2A3 construct under shear. Sequence of simulation snapshots that represents the molecular steps of A2 extension under shear. The sets of amino-acids forming in the native state the β strands $\beta_{1}$, $\beta_{4}$, $\beta_{5}$, $\beta_{6}$ are represented with different colors. The cleavage site ${}^{111}$Tyr is explicitly represented.

**Figure 7 polymers-13-03912-f007:**
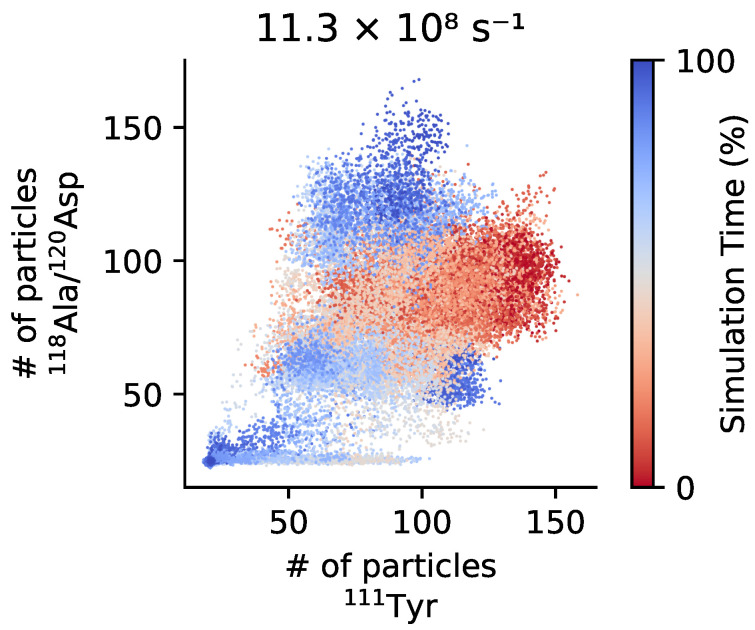
Exposure of the cleavage and the recognition sites of A2 domain under shear. The trajectories of shear unfolding ($\dot{\gamma }=11.3\times 10^{8}\;s^{-1}$) are projected as a scatter plot on the 2D space of the coordination number of ^111^Tyr (cleavage site) and that of ^118^Ala/^120^Asp (recognition site). The sequential exposure of the recognition site before the cleavage site can be appreciated. Trajectories are colored according to progress from red (beginning) to blue (ending).

**Figure 8 polymers-13-03912-f008:**
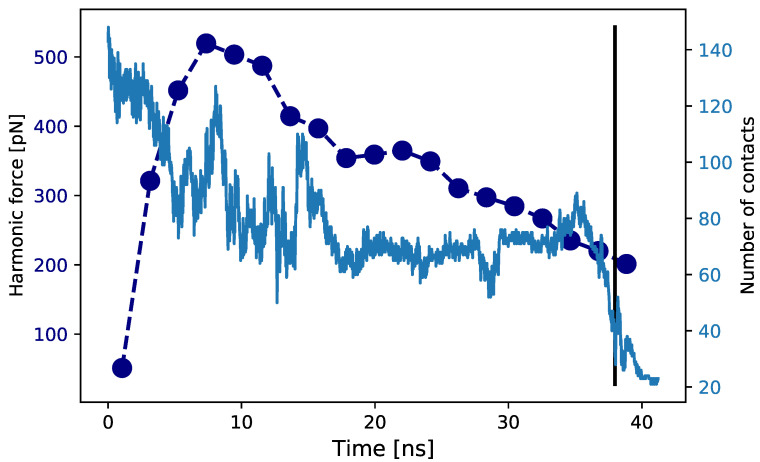
Tensile force. Time evolution of the mechanical force acting at the A2 terminus connected to the $L_{12}$ linker (dark blue). The force is estimated as a moving average over a time window of 2.1 ns. We also report the time evolution of the coordination number of ${}^{111}$Tyr (light blue) until the full exposure criterion is reached, as shown by the vertical line at 38 ns.

**Figure 9 polymers-13-03912-f009:**
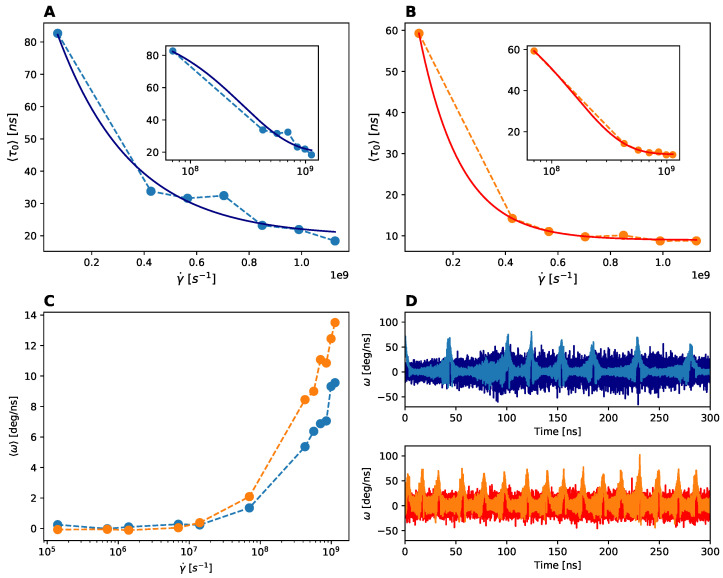
Rotational dynamics. Panels (**A**,**B**). Characteristic rotational time of the A1 (left panel) and A3 (right panel) domains with respect to the central A2 domain as a function of the simulated shear rates $\dot{\gamma }$. In the inset the data are represented in log-x scale for shear rates values $\dot{\gamma }>10^{7}\;s^{-1}$. Panel (**C**). Angular velocity computed for the A1 (blue) and A3 (orange) domain rotation relative to A2 as a function of the shear rate. Panel (**D**). Time evolution of the angular velocity for the A1 (top) and A3 (bottom) domains, for two shear rates $\dot{\gamma }=8.5\times 10^{8}\;s^{-1}$ (light blue and orange) and $\dot{\gamma }=1.4\times 10^{5}\;s^{-1}$ (dark blue and red).

**Figure 10 polymers-13-03912-f010:**
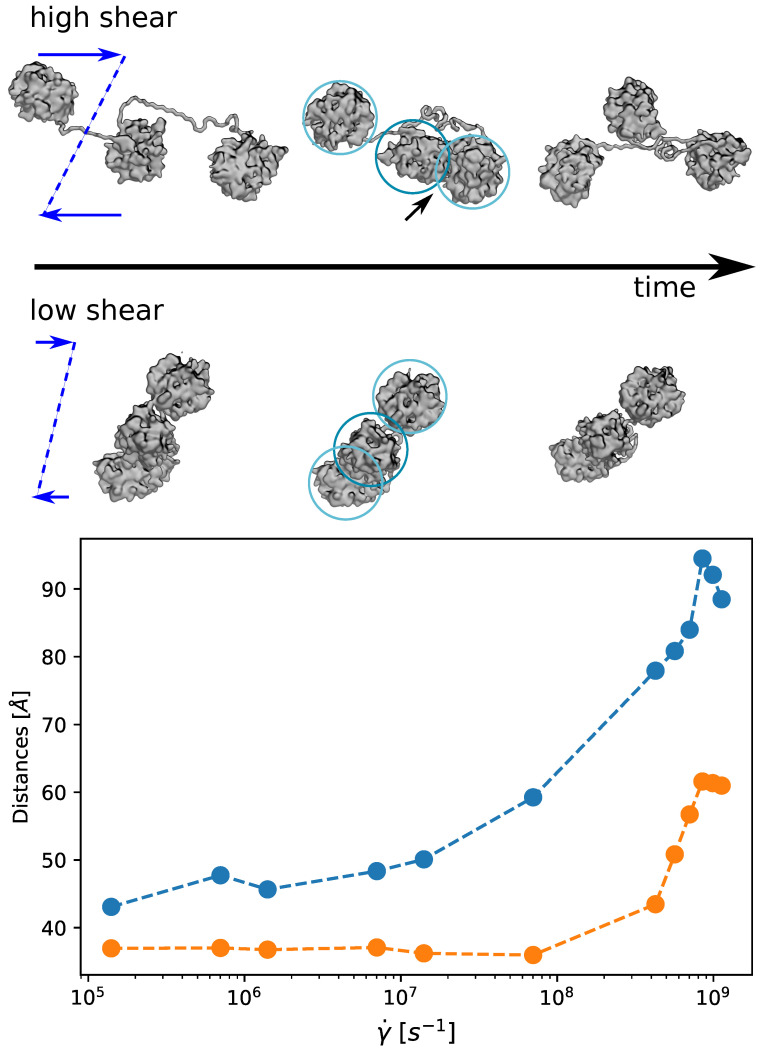
Elongation/Collapse dynamics. Top. Molecular representation of the A1A2A3 system moving in the shear flow at high and low shear rates. At low shear rate the three domains stay collapsed, while at high shear rate they cyclically extend. Bottom. Average inter-domain distances for A1A2 (blue) and A1A3 (orange) as a function of the shear rate.

## Data Availability

The data presented in this study are available on request from the corresponding author.

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
