# Peer review of "Exposure of Von Willebrand Factor Cleavage Site in A1A2A3-Fragment under Extreme Hydrodynamic Shear"

_polymers, 2021, doi:10.3390/polym13223912_

Round 1
Reviewer 1 Report
This paper presents a computational study of hydrodynamics of vWf. The authors first describe the simulation methods and analysis techniques, then they presents and analyses the results concerning the force and shear induced unfolding of vWf.
The paper is well-written, the quality of research is excellent, the results are novel and promising, and the topic is of interest to the readership of Polymers.
I recommend it for publication.
Author Response
We thank the reviewer for the positive feedback.
Reviewer 2 Report
The study by Dr. Languin-Cattoen applies a multiscale modeling of the non-equilibrium unfolding dynamics of the A1A2A3 domain of the VonWillebrand Factor protein under different shear rates. The results are interesting. Here, I list the following questions for the author to address:
- Can the author clarify in the main text that the coordination number is the same quantity as the neighbor backbone atoms within 10 Å of 111Tyr in Page 7/21? This term may be confused with the number of contacts (another variable defined in the main text).
- In fig. 9, why the time autocorrelation function (which is normalized to 1.0) isn’t computed here instead?
- What is the unit of the unfolding rate in y axis of Figs. 2B and 3B?
Minor comment
- Would put the beginning and ending structures alongside the Fig. 2 and 3 be helpful?
- Would a color bar in Figs. 2, 3 and 7 be helpful in showing the migration of points as time elapses, from red (t=0) to white and white to blue?
- In Fig. 10 caption, “At low shear rate the three domains stays” should be “At low shear rate the three domains stay”.
Author Response
We thank the reviewer for the comments.
- We have clearly specified in the text (p.g. 7) that the number of backbone atoms around Tyr111 (or the cleavage residue) is referred to in the text and figures captions as coordination number: "[...] this variable measures how the residue is {\it buried} in the protein matrix before getting exposed to the solvent, and referred as coordination number."
- In figure 9 we reported the characteristic times extracted from the time correlation functions. This allows to observe the change of relaxation times as a function of the shear rate.
- The unfolding rate now is presented as the inverse of the unfolding time, and the proper unit (nanosecond-1) is indicated in the figure. The captions have been changed accordingly.
Minor points:
- The structures of the folded and extended A2 domain under force are indeed reported already in the figure 3.
- We have improved the figures adding a colour bar as suggested by the referee.
- The caption of figure 10 has been corrected.